# Effects of a Multidimensional Exercise and Mindfulness Approach Targeting Physical, Psychological, and Functional Outcomes: Protocol for the BACKFIT Randomized Controlled Trial with an Active Control Group

**DOI:** 10.3390/healthcare13162065

**Published:** 2025-08-20

**Authors:** Belén Donoso, Gavriella Tsiarleston, Yolanda Castellote-Caballero, Alba Villegas-Fuentes, Yolanda María Gil-Gutiérrez, José Enrique Fernández-Álvarez, Santiago Montes, Manuel Delgado-Fernández, Antonio Manuel Mesa-Ruíz, Pablo Molina-García, Rocío Pozuelo-Calvo, Miguel David Membrilla-Mesa, Víctor Segura-Jiménez

**Affiliations:** 1Department of Psychology, Faculty of Education Sciences and Psychology, University of Córdoba, 14005 Cordoba, Spain; belen.donoso@uco.es; 2Physical Activity for Health Promotion (PA-HELP) Research Group, Department of Physical Education and Sports, Faculty of Sport Sciences, University of Granada, 18071 Granada, Spain; albavillegas1994@gmail.com (A.V.-F.); josenrifdz10@gmail.com (J.E.F.-Á.); manueldf@ugr.es (M.D.-F.); 3Instituto de Investigación Biosanitaria ibs.GRANADA, 18012 Granada, Spain; yoligilgu@gmail.com (Y.M.G.-G.); antoniom.mesa.sspa@juntadeandalucia.es (A.M.M.-R.); vsegura@ibsgranada.es (V.S.-J.); 4UGC Medicina Física y Rehabilitación, Hospital Universitario Virgen de las Nieves, 18013 Granada, Spain; pablomolinag5@gmail.com (P.M.-G.); rocio_pozuelo@hotmail.com (R.P.-C.); mmembrilla@gmail.com (M.D.M.-M.); 5Department of Health Sciences, University of Jaén, 23071 Jaén, Spain; mycastel@ujaen.es; 6Faculty of Health Sciences, University of the Middle Atlantic, 35017 Las Palmas de Gran Canaria, Spain; 7Department of Physiotherapy, National School of Higher Studies UNAM, Leon 37684, Mexico; santiagojr96@gmail.com; 8Department of Radiology and Physical Medicine, Faculty of Medicine, University of Granada, 18071 Granada, Spain; 9GALENO Research Group, Department of Physical Education, Faculty of Education Sciences, University of Cádiz, 11003 Cadiz, Spain; 10Instituto de Investigación e Innovación Biomédica de Cádiz (INiBICA), 11003 Cadiz, Spain

**Keywords:** physical fitness, impairment, mindfulness, non-specific chronic low back pain, exercise, resistance training, randomized controlled trial

## Abstract

**Introduction:** Chronic primary low back pain (CPLBP) is a prevalent condition in primary care and a leading cause of disability and absenteeism worldwide. Multidimensional approaches may be necessary to achieve physical and mental health benefits in individuals with CPLBP. **Objective:** The BACKFIT randomized controlled trial aims to evaluate the effectiveness of a multidimensional intervention—combining supervised exercise and mindfulness—on pain, physical fitness, mental health, and functional outcomes in individuals with CPLBP. **Hypothesis:** Both the supervised exercise program focused on motor control and trunk muscle strength (IG1) and the multidimensional intervention combining supervised exercise with mindfulness training (IG2) are expected to produce significant health improvements in individuals with CPLBP. It is further hypothesized that IG2 will yield greater improvements compared to IG1, both immediately post-intervention and at the two-month follow-up. **Design:** Randomized controlled trial. **Setting:** Virgen de las Nieves University Hospital, Granada (Spain). **Participants:** 105 individuals. Inclusion criteria: Previously diagnosed with CPLBP, aged ≥18 and ≤65 years, able to read and understand the informed consent, and able to walk, move, and communicate without external assistance. Exclusion criteria: serious lumbar structural disorders, acute or terminal illness, physical injury, mental illness, and medical prescriptions that prevent participation in the study. **Intervention:** Individuals will be randomly assigned to a supervised physical exercise group (2 days per week, 45 min per session), a multidimensional intervention group (same as supervised physical exercise group, and mindfulness 1 day per week, 2.5 h per session) or an active control group (usual care, 2 days per week, 45 min per session). The intervention will last 8 weeks. **Main Outcome Measures:** Primary outcome: pain threshold, perceived acute pain, and disability due to pain. Secondary measures: body composition, muscular fitness, gait parameters, device-measured physical activity and sedentary behavior, self-reported sedentary behavior, quality of life, pain catastrophizing, mental health, sleep duration and quality, and central sensitization. The groups will undergo pre-intervention, post-intervention, and a 2-month follow-up after a detraining period. **Statistical Analysis:** Both per-protocol and intention-to-treat approaches (≥70% attendance) will be used. Program effects will be assessed via one-way ANCOVA for between-group changes in primary and secondary outcomes. **Conclusions:** Given the complex nature of CPLBP, multidimensional approaches are recommended. If effective, this intervention may provide low-cost alternatives for health professionals.

## 1. Introduction

Low back pain (LBP) is one of the most prevalent chronic pain conditions and the leading cause of years lived with disability worldwide [1]. Its high and growing prevalence across diverse populations poses a major global health challenge, particularly in terms of effective prevention and long-term management [2]. Despite its substantial burden, chronic and non-specific forms of LBP remain insufficiently understood and often suboptimally managed, highlighting the need for evidence-based and individualized therapeutic strategies [3]. Chronic low back pain (CLBP) is defined as pain located between the lower margin of the ribs and the sacral region, lasting more than 12 weeks [4]. It is typically classified as either specific, when a clearly identifiable pathology is present, or non-specific, when no underlying structural cause can be reliably determined. Recently, the World Health Organization (WHO) (2023) [5] introduced the term chronic primary low back pain (CPLBP) to describe persistent or recurrent low back pain lasting more than three months, not attributable to a specific disease (e.g., cancer, infection) or structural abnormality (e.g., fracture, deformity). CPLBP is considered a clinical subset of non-specific chronic low back pain and is often associated with emotional distress and functional impairment. In Spain, the National Health Survey (2023) reported a prevalence of 19.8% for CLBP among individuals aged 15 years and older. This underscores the considerable public health burden of the condition at the national level and reinforces the need for accessible, effective, and evidence-based interventions to mitigate its clinical and societal impact [6].

In the adult population, longer sedentary time has been associated with greater low back pain [7], possibly due to increased stiffness in lower body musculature, impaired blood flow, or poor postural habits derived from prolonged sitting [8,9]. Moreover, individuals with CPLBP often fail to meet the World Health Organization’s physical activity (PA) recommendations (<150 min/week of moderate intensity or <75 min/week of vigorous intensity) and typically engage in lower levels of PA than age-matched healthy peers [10,11,12]. Given that PA contributes to delaying functional decline and improving quality of life, and is closely linked to health parameters in both general and clinical populations [10], it could be particularly relevant to determine, through objective measures such as triaxial accelerometry, the levels of PA and sedentary behavior in individuals with CPLBP. A recent scoping review [13] highlighted the need for greater standardization and methodological rigor in the use of accelerometry in this population, supporting its inclusion in clinical trials focused on PA monitoring.

Evidence-based clinical guidelines for the management of CPLBP support the use of exercise training as a first-line therapy [14,15]. Supervised exercise is considered key for addressing pain and disability. Individuals with CLBP who engage in supervised programs have demonstrated improvements in short- and long-term outcomes such as muscle endurance, pain intensity, functional mobility, flexibility, and quality of life [16]. Although most exercise interventions yield benefits in managing pain and disability in CLBP, effect sizes remain small to moderate, similar to other standard treatments [17,18].

A recent network meta-analysis [19] found that active interventions such as tai chi, yoga, Pilates, sling exercises, and core stabilization were more effective than conventional rehabilitation or no treatment in reducing pain and improving physical function. The authors emphasized the importance of therapies that target core strength and postural control as effective strategies for managing CLBP. Consistently, another network meta-analysis concluded that the most beneficial programs included: (1) at least 1 to 2 sessions per week of Pilates or strength training; (2) sessions lasting less than 60 min focusing on core, strength, or mind-body components; and (3) training durations of 3 to 9 weeks using Pilates or core-based exercises [20].

Additionally, trunk-focused exercise programs have shown positive effects on pain, disability, quality of life, and trunk performance compared to control groups, and superior results in pain and disability when compared to general exercise in individuals with CPLBP [21]. Trunk muscle strength and endurance may also serve as objective indicators in differentiating treatment responses and evaluating the effectiveness of exercise interventions [22]. Furthermore, evidence from young athletic populations supports the clinical relevance of trunk muscle characteristics in LBP. In adolescent gymnasts, an imbalanced flexor-to-extensor strength ratio and reduced lumbar muscle thickness have been associated with CPLBP, suggesting that deficits in trunk muscle function and morphology may represent key targets for rehabilitation [23].

Although recent network meta-analyses have contributed valuable insights into the comparative effectiveness of exercise-based treatments for CLBP, important methodological limitations persist. As highlighted in a previous study [19], considerable heterogeneity exists in terms of intervention type, frequency, intensity, and duration, with many trials implementing very short programs (≤3 weeks) and lacking follow-up measures. The present study aims to address these gaps by combining an 8-week structured exercise intervention focused on trunk muscle strengthening with a 2-month follow-up. This design is expected to provide evidence on the short-term effectiveness and clinical applicability of multidimensional, non-pharmacological interventions in individuals with CPLBP.

Recently, mindfulness has gained recognition within the research and medical communities [24]. Mindfulness meditation might reduce pain and/or the emotional experience associated with it [25,26]. Neuroimaging studies suggest that mindfulness meditation modulates pain by enhancing activity in brain regions involved in cognitive and emotional appraisal (e.g., anterior cingulate cortex, orbitofrontal cortex, right anterior insula), and reducing activation in sensory regions like the thalamus, thus dampening nociceptive signaling to areas such as the somatosensory cortex [26,27]. These effects support mindfulness as a tool for re-evaluating pain experiences. The Mindfulness-Based Stress Reduction (MBSR) program, a structured 8-week intervention, integrates meditation, awareness, and movement practices to improve coping with discomfort and emotional distress. Studies report MBSR benefits for chronic pain, including reduced pain severity, improved mental health, and enhanced quality of life and physical functioning [28,29]. Telehealth-delivered mindfulness has shown promise, yet the impact of in-person programs remains understudied. This study aims to evaluate a supervised, face-to-face mindfulness intervention, hypothesizing enhanced outcomes through therapist support, group interaction, and greater engagement.

Multidisciplinary rehabilitation has been shown to be effective for short- and intermediate-term improvements in pain and function in individuals with chronic musculoskeletal pain [30]. Previous research combining therapeutic exercise and mindfulness reported large effect sizes for functional impairment, pain, kinesiophobia, and physical function, as well as moderate effects for fatigue, anxiety, and depressive symptoms in individuals with fibromyalgia [31]. Compared to usual care, multicomponent treatments demonstrated short-term (up to 3 months) positive effects in these individuals. However, multidimensional approaches have been scarcely studied in individuals with CPLBP. A recent study involving an exercise program and patient education (self-management) highlighted the great potential of these approaches in individuals with CPLBP [32]. Furthermore, an intervention combining exercise and meditation was effective at decreasing self-reported perceptions of LBP from baseline in this population [33]. Similarly, a previous study evaluated an 8-week online program combining MBSR and exercise in adults with chronic pain, reporting high adherence and participant satisfaction [34]. Although no significant differences were found between groups, both groups showed improvements in pain, disability, and pain catastrophizing, supporting the feasibility of integrating mindfulness and exercise in this context. However, few studies have explored multidimensional interventions that combine supervised exercise and mindfulness, particularly in a face-to-face format. The present clinical trial aims to contribute to this emerging field by evaluating the effects of a structured, in-person intervention on a range of physical and psychological outcomes in individuals with CPLBP.

With the aim of understanding the effects and magnitude of these multidimensional programs on health outcomes in individuals with CPLBP, the BACKFIT randomized controlled trial was initiated. The main objective is to determine the effectiveness of a multidimensional program (combining supervised physical exercise and mindfulness), compared to supervised physical exercise alone and an active control group (usual care), on pain and disability (primary outcomes), body composition, muscular fitness, gait parameters, device-measured PA and sedentary behavior, self-reported sedentary behavior, quality of life, pain catastrophizing, mental health, sleep duration and quality, and central sensitization (secondary outcomes) in individuals with CPLBP.

## 2. Methods

### 2.1. Study Design

This study will be a randomized controlled trial (RCT), registered on ClinicalTrials.gov (NCT05443880) with three arms: intervention group 1 (supervised physical exercise), intervention group 2 (supervised physical exercise and mindfulness), and an active control group (usual care). The study will compare the outcomes between these groups in individuals with CPLBP. The study design aligns with the Consolidated Standards of Reporting Trials (CONSORT) guidelines [32]. The Standard Protocol Items: Recommendations for Interventional Trials (SPIRIT) guidelines [35,36] are followed to present this protocol, as detailed in Appendix A. The interventions adhere to the Template for Intervention Description and Replication (TIDieR) guidelines [37] for clinical trials, as outlined in the Appendix A. A study flowchart is provided in Table 1 and Figure 1.

### 2.2. Participants

This RCT, based on a supervised exercise program and mindfulness intervention, was prescribed and planned by physical therapists and sports science professionals, and it will be conducted in Granada (southeastern Spain). Based on previous literature [22], with a statistical power of 0.80, an alpha error of 0.05, and an effect size of 0.70, a sample size of 26 participants per group is sufficient to find differences in pain after the intervention. Considering a dropout rate of approximately 20–30%, as reported in similar studies involving pain populations [39], the total sample size will be set at 105 participants, randomly assigned to intervention group 1 (IG1, *n* = 35), intervention group 2 (IG2, *n* = 35) and control group (CG, *n* = 35). The inclusion and exclusion criteria are described in Table 2.

### 2.3. Recruitment and Randomization

Participant recruitment will take place at the Physical Medicine and Rehabilitation Service of Virgen de las Nieves University Hospital in Granada (belonging to the Andalusian Health Service). Individuals referred to the Rehabilitation Unit by their medical specialist and placed on a waiting list will receive information about the possibility of participating in this RCT via email or a telephone call. The diagnosis of CPLBP will be conducted by medical specialists in Physical and Rehabilitation Medicine, with advanced training and extensive clinical experience in musculoskeletal disorders. Subsequently, those initially interested in participating will be invited to an informational session where the study will be explained, and any concerns that may arise will be addressed. The estimated recruitment period for selecting participants will be from 1 to 3 months. Additionally, all interested individuals will be provided with written information about the study.

Individuals who agree to enroll in the study will be asked to read and sign a written informed consent that allows them to withdraw at any time. A unique code will be assigned to each participant to ensure their anonymity. On the first evaluation day, participants will report to the Multidisciplinary Laboratory in the Faculty of Sport Sciences (Granada) and will complete the following assessments: clinical information survey, blood pressure, resting heart rate, body composition, pain threshold, physical fitness, and gait parameters. The initial survey (anamnesis) will be conducted through face-to-face interviews by trained staff in order to collect data on sociodemographic characteristics (age, marital status, educational level, and current occupation), medical records and comorbidities, medicine consumption, and health habits (tobacco, alcohol, etc.). The Spanish version of the PAR-Q (C-AAF) [40] will be used to detect individuals who should undergo a medical examination before doing any type of PA. Individuals will not be required to perform any additional physical exercise or receive treatment for CPLBP upon starting this RCT. Participants will receive an accelerometer and several health-related questionnaires to complete at home on the same day of the evaluation. Participants will be asked to return the accelerometer and questionnaires nine days later. As a reward, they will receive a report for each assessment they attend.

Once the baseline evaluations have been completed, participants will be randomized into different groups to reduce the risk of bias during the assessment. A computer-generated randomization sequence stratified by two factors (age at study entry and sex) will allocate participants to each group. This randomization sequence will be created by a blinded researcher external to the project using computer software. The allocation will be concealed in a password-protected computer file.

The groups will undergo pre-test (before the intervention), post-test (after each intervention), and follow-up (after an 8-week detraining period) measurements.

### 2.4. Intervention

The intervention program will be conducted face-to-face in groups at the Physical Medicine and Rehabilitation Service. Each session will have a maximum of 17 participants. To accommodate participants’ availability, all intervention groups will offer morning and afternoon sessions. The physical exercise program will be supervised and guided, and all sessions will be conducted under the direct supervision of licensed (MD and Ph.D.) physical therapists and sports sciences specialists present physically. Mindfulness sessions will be taught by a professional accredited by Brown University who will not participate in any assessment phase. All the professionals will have experience working with this population. The supervised exercise program will be applied by adapting rehabilitation exercise methods used in previous literature [32,35] and will be analyzed with the Consensus on Exercise Reporting Template (CERT) [36]. This guide consists of 16 items that describe information about the performance of an exercise intervention program. This facilitates its development, guidance, evaluation, interpretation, and clinical use. The program will report the FITT exercise principles (frequency, intensity, time, and type) as a detailed and high-quality physical exercise intervention.

The program will follow National Strength and Conditioning Association [37] and American College of Sports Medicine [41] recommendations, suggesting 6 to 12 repetitions per muscle-strengthening PA at moderate-to-vigorous intensity, equivalent to >5 on the Rating of Perceived Exertion (RPE) scale (1 ‘no effort’ to 10 ‘maximal effort’) [42].

The IG1 (supervised exercise program) will undergo an exercise program twice per week (45 min per session) for 8 weeks. Each session will include a warm-up (5 min), muscle-strengthening exercises (35 min), and a cool-down (5 min, stretching exercises). The program will focus on core muscles, starting with low-intensity isometric contraction for trunk stabilization and mobility exercises, then increasing intensity with functional tasks. Exercise intensity will be moderate-to-vigorous, assessed via the RPE scale [42], ensuring no pain is felt. This scale will be visible to participants during sessions. Each exercise will last 60 s, followed by 60 s rest. Execution speed will be individualized to each participant’s physical condition and tolerance. Exercises will follow a fixed order with progressive difficulty, as described in previous literature [32,43]. This progression will be structured according to the following phases [32]. (i) *Phase 1* (3 sessions, ≥5 RPE): mobility, isometric, and motor control exercises that will include full body, with special emphasis on the core, particularly isometric contraction of the transversus abdominis muscle. Participants will be instructed to place their hands on the lower abdomen during the exercises to perceive the contraction through kinesthetic feedback and facilitate correct transversus abdominis activation (ii) *Phase 2* (4 sessions, ≥5 RPE): co-contraction and functional tasks involving deep trunk muscles. (iii) *Phase 3* (4 sessions, ≥6 RPE): functional task with greater difficulty or intensity, including load. (iv) *Phase 4* (5 sessions, ≥6 RPE): functional task performed on an unstable surface.

A detailed description of the exercise intervention is shown in Table 3 and Table 4. Participants will receive real-time feedback on posture correction and examples for exercise familiarization. Strategies to improve exercise adherence will include: (i) a positive and dynamic session environment; (ii) an attendance control system to minimize dropouts.

The IG2 (multidimensional program) will follow the same exercise protocol as IG1 (twice per week, 45 min per session) and also participate in a mindfulness program once per week (2.5 h per session) for 8 weeks. The MBSR program will follow Jon Kabat-Zinn’s protocol [44]. Each session will include a topic presentation, group dialogue and exploration (using appreciative inquiry), and a mindfulness practice. Participants will receive workbooks, guided meditation audios, and instructions for home practice. Details are provided in Table 5.

**Table 5 healthcare-13-02065-t005:** Overview of the Mindfulness-Based Stress Reduction (MBSR) program.

Session	Content Description	Class Details	Session	Content Description	Class Details
1	- Brief history of the MBSR program. Definition and scientific evidence. - What mindfulness is and is not. - Example of a standard mindfulness practice. - Guided reflection: “What brought you here?” - Introduction of participants and instructor. - Raisin meditation. - Body scan. - Presentation of home practice. - Closing practice.	Introduction session 2.5 h Individual practice Face-to-face Chairs	2	- Opening practice. - Standing yoga postures. - Body scan. - Group dialogue (small and large group). - 9-point exercise: conditioning and creative response. - Seated meditation with attentional focus. - Presentation of home practice. - Closing practice.	Introduction session 2.5 h Individual and group practice Face-to-face Chairs, mats
3	- Opening practice. - Focused attention meditation. - Floor yoga postures. - Dialogue on personal experience. - Review of the Pleasant Events Calendar (awareness of bodily sensations, thoughts, emotions). - Presentation of home practice. - Closing practice.	2.5 h Individual and group practice Face-to-face Chairs, mats, notebook	4	- Opening practice. - Standing yoga postures. - Seated meditation focusing on unpleasant experiences (pendulum method). - Dialogue and reflection. - Review of the unpleasant events calendar and link with stress. - Explanation of stress physiology and reactivity.Presentation of home practice. - Closing practice.	2.5 h Individual and group practice Face-to-face Chairs, mats, notebook
5	- Opening practice. - Standing yoga postures. - Full meditation alternating focus points (open awareness). - Dialogue on progress, challenges, and expectations. - Introduction to the habit loop and the mindful pause. - Presentation of home practice. - Closing practice.	2.5 h Individual and group practice Face-to-face Chairs, mats	6	- Opening practice. - Standing yoga postures. - Extended seated meditation, with more silence and less guidance. - Dialogue on practice engagement and outcomes. - Guided reflection on a difficult communication situation and interactive dialogue. - Presentation of home practice. - Closing practice.	2.5 h Individual and group practice Face-to-face Chairs, mats
7	- Opening practice. - Movement practice: simplicity and boundary exploration. - Seated meditation (extended silence). - Dialogue on experience. - Reflection on relationship with technology and its impact. - Presentation of home practice. - Closing practice.	2.5 h Individual and group practice Face-to-face Chairs, mats	8	- Opening practice. - Body scan. - Group dialogue on home practice. - Writing a letter to the “future self”. - Resources for long-term practice (“the rest of your life”). - Sharing of final reflections and program integration. - Closing practice.	2.5 h Individual and group practice Face-to-face Chairs, mats, notebook

The Mindfulness-Based Stress Reduction program is based on previous evidence [44,45]. Each session includes a topic presentation, group dialogue and exploration, and a mindfulness practice.

The CG will receive the usual care provided by the Physical Medicine and Rehabilitation Service. Participants will follow the same session structure described above twice per week (45 min per session) for 8 weeks. All participants will perform 1 set of 10 repetitions or 60 s per exercise, followed by 60 s of rest. Execution speed will be individualized. Intensity will be assessed with an RPE scale (≥5 RPE). Sessions will include: warm-up (5 min: postural awareness, diaphragmatic breathing, costal breathing, transversus abdominis contraction, pelvic swing); main activity (35 min: lower/upper abdominis exercise, hip abduction with knee extension, cat-camel exercise, alternate arm/leg elevation in quadruped, bird-dog); and cool-down (5 min: lumbosacral, psoas, hamstring, and pyramidal muscle stretching, bird-dog, and lumbosacral stretching on the floor). All of them will be performed in supine or quadrupedal position. Although active, this protocol reflects hospital standard rehabilitation care. Including it in the control group meets ethical and organizational requirements, ensuring participants are not deprived of care, allowing a more rigorous comparison with the multidimensional intervention.

This program allows adaptations for practical evidence-based aspects, such as performing an alternative exercise targeting the same muscle group without causing pain or changing the plane of movement. Adverse events, effects, or health issues attributable to the testing or intervention sessions, along with their severity and resolution, will be recorded by a researcher responsible for auditing the assessment team. If an adverse event prevents participation, temporary or permanent exclusion will be discussed and documented. Temporary exclusions require medical clearance for return.

This study was approved by the Clinical Research Ethics Committee of Granada, Government of Andalusia, Spain (code: 2255-N-21). It will be conducted following the ethical principles of the Declaration of Helsinki (1964 and later amendments). No formal audit is scheduled for this clinical trial.

## 3. Results

### 3.1. Primary Outcome Measures

#### 3.1.1. Pain-Related Measures

##### Pressure Pain Threshold

Algometry will be used to measure pain threshold using a hand-held standard pressure algometer (FPK 20, Wagner Instruments, Greenwich, CT, USA) [46]. Two measurements will be conducted twice, bilaterally, lying in the prone position. Each measure will be assessed in the middle side between L1–L5, 5 cm apart from the lumbar spinal processes [47,48,49]. Pressure pain threshold (PPT) will be defined as the minimum pressure perceived as painful by the participant, who will be instructed to say “stop” upon the onset of pain sensation [49]. To minimize the risk of temporal summation, PPT measurements will be taken twice at each site with a 30-s interval between assessments, and the average value will be used for data analysis. Static measures of PPT, or the point where stimulation just becomes painful, have demonstrated good to excellent reliability [50].

##### Perceived Acute Pain

A visual analogue scale (VAS) will be used to measure subjective changes in back pain intensity and unpleasantness. Testing will be performed at arrival and immediately after performing each physical fitness test during assessment [51]

A numerical rating scale (NRS) will be used to assess the pain before and after each intervention session. It is a 0–10-point scale, where 0 indicates “no pain” and 10, the “worst imaginable pain” [52].

##### Pain Catastrophizing

Pain Catastrophizing Scale (PCS) [53] will be used to assess the pain catastrophizing. It is a widely used and validated tool assessing three dimensions: rumination, magnification, and helplessness associated with pain [54]. This scale consists of 13 items, with scores for each question ranging from 0 to 4. The theoretical range of the instrument spans from 13 to 52, with higher scores reflecting a greater tendency to catastrophize pain symptoms.

##### Disability Due to Pain

The Oswestry Disability Index (ODI) will be used to measure pain disability. It is a valid and reliable self-administered questionnaire, specific for LBP, that measures limitations in activities of daily living [55]. It consists of 10 questions that are rated from 0 to 5 (from least to most limited). The total score (0–100%) is calculated by dividing the sum of the individual item scores by the maximum possible score and multiplying by 100. The interpretation of the results is as follows: 0–20% indicates minimal functional limitation; 20–40%, moderate limitation; 40–60%, severe limitation; 60–80%, disability; and above 80%, maximum functional limitation.

### 3.2. Secondary Outcome Measures

#### 3.2.1. Body Composition

Weight (kg), fat (%), skeletal muscle mass (kg), and fat mass (kg) will be measured by 8-point tactile-electrode bioimpedance (InBody R20, Biospace Gateshead, Kent, UK). The validity and reliability of this instrument have been previously reported [56]. Height will be assessed with a height rod (Seca 22). Moreover, waist [57] and neck circumferences [58] will be measured with a measuring tape (Seca 201).

#### 3.2.2. Muscular Fitness

The Biering–Sørensen test [59] will be used to evaluate back-extensor muscle strength. This test is commonly used to assess the endurance of the back and hip muscles, which are inversely associated with the presence and severity of LBP [60]. It is a trunk holding test in an antigravity prone position. The participant is placed in a prone position with the iliac crest at the edge of the stretcher. Two straps will be attached to the waist and the knees of the participant. At the investigator’s signal, the participant should try to keep the back in a horizontal position with hands at the side of the head as long as possible. Participants will perform only one trial, and the time will be recorded in seconds.

The Prone Bridging test [61] will be used to measure back-flexor muscle strength. It is a reliable isometric holding test in prone position with elbows under the glenohumeral joint, humeri joint perpendicular to the floor; head aligned with the cervical spine; trunk and pelvic floor in neutral position; knees extended and toes contacting the ground. Participants will keep this position as long as possible. When this position is maintained for 4 min, the test will be stopped. The test has demonstrated good reliability [62] and may serve as a useful indicator of core muscle endurance. Its performance time is inversely associated with the presence and intensity of LBP [63].

The 30-sec chair stand test will be used to evaluate the quadriceps, hamstrings, and gluteus muscle strength. It has shown good reliability in previous research with older adults and chronic-pain individuals [64]. Participants will be seated with their back straight on the chair, feet flat on the floor, and arms crossed and held against the chest. From this position, participants will rise to a full standing position and then return to the sitting position, as many times as possible in 30 s.

Hand dynamometry (5101 TKK handgrip dynamometer) will be used to measure upper body muscle strength. Participants will stand with shoulder abduction ~10 cm separated from the body, extended elbow, and the wrist in neutral position. The participant will press the dynamometer with maximum muscle contraction for 3–5 s without any additional body movement. The best score from both hands will be averaged. Previous research has shown a significant association between handgrip strength and CLBP [61] in these individuals. This test has shown good reliability in older adults [65]

#### 3.2.3. Motor Agility

The 8-foot up-and-go test will be used to measure motor agility. Participants had to stand up from a chair, walk 2.44 m to and around a cone, and return to the chair in the shortest period of time. The best time from the two trials will be recorded. The 8-foot up-and-go test is a reliable and validated tool commonly used to assess motor agility in a safe manner [64].

#### 3.2.4. Gait Parameters

The high-density photoelectric cell platform, Optogait™ (Optogait; Microgate, Bolzano, Italy), will be used to evaluate spatiotemporal gait parameters, such as contact time, cadence, gait velocity, stance, support time, and stride length. Optogait™ has been demonstrated to be a valid and reliable method to assess gait parameters [66,67]. Participants will be informed to walk comfortably and repeat the circuit until a total of 200 steps is registered.

#### 3.2.5. Device-Measured Physical Activity and Sedentary Behaviour

The triaxial ActiGraph GT3X+ accelerometer (Actigraph, Inc., Fort Walton Beach, FL, USA) will be used to objectively assess PA and sedentary time. Participants will wear the accelerometer for 9 days (24 h a day) except during water activities. Days of device receipt and return (non-complete days) will be excluded from the analyses [68]. A minimum of 10 h of wear time per day and a total of 7 days with valid data will be required [69]. The accelerometer will be worn on the dorsal surface between the ulnar and radial styloid of the non-dominant wrist, tied with an elastic strap. It is suggested that wrist placement can help reduce selection bias and present minimal discomfort during sleep [70]. Accelerometer wear-time will be calculated by subtracting sleep and nap time (through a diary where participants reported the time they went to bed and the time they woke up) from each day. Participants will also record daily shower and/or water activities time in a diary [69,71,72]. Total time in sedentary behavior, light and moderate-to-vigorous PA (min/day) will be calculated based upon the recommended vector magnitude cut point [69,72].

#### 3.2.6. Self-Reported Sedentary Behavior

The Spanish version of the Sedentary Behavior Questionnaire (SBQ-S) [73] will be used to assess self-reported sedentary behavior. Participants will report how long they spent in 11 different sedentary behaviors. Response options are none, 15 min or less, 30 min, 1 h, 2 h, 3 h, 4 h, 5 h, or 6 h or more. This questionnaire has demonstrated adequate reliability and validity in adult populations and has also been previously employed in individuals with chronic pain [73].

#### 3.2.7. Quality of Life and Mental Health

The 36-item Short-Form Health Survey (SF-36) [74] will be used to evaluate health-related quality of life. It contains 36 items grouped into 8 dimensions. The scores range from 0 to 100 in every dimension, where higher scores indicate better health. It has demonstrated appropriate validity [74].

The Beck Depression Inventory-II (BDI-II) [75] will be used to assess depression severity. It contains 21 items assessing depressive symptoms, ranging from 0 (not present) to 3 (severe), according to how they have been feeling during the previous two weeks. The range score is 0–63: minimal (0–13); mild (14–19); moderate (20–28); and severe (29–63).

The State-Trait Anxiety Inventory-I (STAI) [76] will be used to evaluate the level of current anxiety. It is a 20-item self-administered questionnaire in which individuals are asked to indicate the score from 0 to 3 that best indicates how they feel at the time they are filling in the questionnaire. The range of scores is 20–80: no anxiety (≤20), mild (21–39), moderate (40–59), and severe anxiety (60–80).

#### 3.2.8. Sleep Quality

The Pittsburgh Sleep Quality Index (PSQI) will be used to assess the sleep duration and quality [77]. The PSQI is 19 items composed of 7 subscales, where scores range from 0 to 3 (global score 0–21). Higher scores indicate poor sleep quality. This instrument has been previously used in studies involving individuals with chronic pain [78].

#### 3.2.9. Central Sensitization

The Central Sensitization Inventory (CSI) will be used to evaluate symptoms related to central sensitization (CS) and central sensitivity syndromes (CSS) [79,80]. It contains 25 statements related to current health symptoms. Each of these items is measured on a 5-point temporal Likert scale. The score range is 0–100: subclinical (0 to 29), mild (30 to 39), moderate (40 to 49), severe (50 to 59), and extreme (60 to 100). CSI has demonstrated good reliability, internal consistency, and construct validity [80].

#### 3.2.10. Rate of Perceived Exertion

RPE based on the Borg CR-10 scale will be used to assess training intensity. It is a 10-point scale ranging from 0 (“nothing at all”) to 10 (“very, very strong”) [81]. It represents a valid and moderately reliable tool for monitoring exercise intensity in chronic pain populations [82]. At the end of each session, the participants will determine the subjective intensity of the session. The CR-10 will also be used after performing each physical fitness test during assessments.

## 4. Statistical Analysis

Since this study aims to determine the potential efficacy and effectiveness of supervised exercise programs and mindfulness, statistical analysis will be performed using both per-protocol and intention-to-treat approaches (with ≥70% attendance required for inclusion). The effects of the intervention will be assessed as between-groups differences in changes from baseline to post-intervention in primary and secondary outcomes using one-way analysis of covariance (ANCOVA). The mean change (post-intervention minus baseline values) will be used as the dependent variable, the group as the fixed factor, and the baseline value of the outcome as a covariate. The same approach will be applied to evaluate the persistence of changes at follow-up. Given the repeated-measures design, linear mixed models will also be used to examine group-by-time interactions across all time points. This approach accommodates missing data under the missing-at-random assumption and accounts for within-subject correlation, providing a more robust framework than ANCOVA for longitudinal analyses. All analyses will be adjusted for potential confounding variables that are not well balanced at baseline. Sex will be used as a potential covariate in all analyses to explore whether the intervention effects differ between men and women, and whenever possible, the sample will be stratified by sex. Cohen’s d will be calculated to determine standardized effect sizes. To control for type I error due to multiple comparisons, the Bonferroni correction will be applied when conducting post hoc analyses or when evaluating multiple outcome variables. Statistical significance will be set at α = 0.05. All analyses will be performed using the Statistical Package for the Social Sciences (SPSS) for Windows, Version 27.0. (IBM Corp., Armonk, NY, USA).

## 5. Discussion

This paper outlines the protocol developed by a multidisciplinary team of experts in Physical Therapy and Sports Sciences, aimed at determining the effects of a supervised exercise intervention and mindfulness (i.e., a multidimensional approach) on health outcomes in individuals with CPLBP, an area that remains relatively underexplored. In line with the latest WHO guidelines, which highlight the complexity and heterogeneity of mechanisms involved in CPLBP, multimodal and multidisciplinary interventions are recommended over isolated treatments for individuals who do not respond adequately to single modalities. These approaches not only aim to enhance therapeutic effectiveness but may also contribute to reducing future clinical costs and long-term healthcare needs [32]. Previous studies support the use of integrative, non-pharmacological strategies to address both the physical and psychological dimensions of chronic pain [19,24,83]. Based on this rationale, the present protocol combines supervised physical exercise and mindfulness to simultaneously address multiple health-related outcomes. It is expected that both intervention groups will show improvements in pain, disability, trunk muscle function, physical fitness, gait, PA levels, sedentary behavior, sleep duration and quality, mental health, and quality of life compared to the standard exercise care group. Furthermore, greater improvements are anticipated in the group receiving the combined intervention (IG2), particularly in outcomes related to pain, psychological well-being (e.g., pain catastrophizing, mental health), sleep quality, and central sensitization.

Previous research has shown that individuals with CPLBP often experience persistent pain and inflammation [19]. These conditions may lead to reduced muscle strength and functional limitations, as nociceptive signals from the spine can inhibit neuromuscular control mechanisms [19]. Interventions such as physical exercise aimed at improving spinal stability, movement control, and neuromuscular connectivity may help to increase the pressure pain threshold in this population. These benefits are expected given that CPLBP is associated with diminished neural drive to key stabilizing muscles, such as the multifidus and erector spinae [19,84]. Consequently, motor control is considered a relevant physiological mechanism in CPLBP, as impaired neural drive alters proprioceptive feedback and postural control [19]. In the present protocol, one of the four phases of the physical exercise program targets proprioception and balance. These components are expected to enhance motor control and contribute to improved spinal stability and functional movement.

With moderate certainty, structured exercise programs are likely to reduce pain and functional limitations in adults and older individuals with CPLBP [18]. Trunk-focused exercise [21,23], motor control, core stability, and Pilates programs [85] have all been shown to improve pain and disability in this population.

Given that CPLBP often leads to psychological deterioration and a decline in health-related quality of life, mindfulness-based interventions may be particularly appropriate to improve both mental health and overall quality of life in this population [33,86,87,88,89,90]. Among these, MBSR is a non-pharmacological therapy that integrates physical and psychological components and has shown promising results in the management of chronic pain [24]. This approach encourages individuals to perceive pain without judgment, helping them dissociate the sensory experience of pain from maladaptive emotional and cognitive responses, and fostering acceptance rather than avoidance [24]. According to a previous study [83], mindfulness may attenuate pain by engaging the prefrontal cortex, associated with pain evaluation, and modulating thalamic activity, thereby inhibiting ascending nociceptive input. In addition to its effects on pain processing, mindfulness meditation has also been associated with improvements in mental health outcomes [83]. It appears to reduce emotional reactivity and enhance cognitive control; both mechanisms are linked to improved mood regulation and reduced anxiety. These effects are thought to be mediated by increased activity in the prefrontal cortex and reduced activation of limbic regions, contributing to greater emotional resilience and explaining the observed reductions in anxiety, depression, and pain catastrophizing among mindfulness practitioners.

Given the well-established association between CPLBP and psychological distress, the inclusion of validated instruments to assess mental health outcomes, is considered essential. Psychological factors such as anxiety and depression are highly prevalent in individuals with CLBP and influence the intensity of symptoms, the level of disability, and the overall response to treatment [1]. Therefore, the present protocol includes the Beck Depression Inventory-II (BDI-II) and the State-Trait Anxiety Inventory (STAI) to measure symptom changes in depression and anxiety, respectively. These measures are particularly appropriate within the context of a mindfulness-based intervention, which seeks to enhance emotional regulation and psychological well-being by reducing maladaptive cognitive and emotional responses to pain.

The growing prevalence of chronic diseases in adults is closely linked to sedentary lifestyles and a lack of regular physical activity. In this context, developing effective rehabilitation strategies for individuals with CPLBP could help reduce the burden on healthcare systems. If proven effective, this intervention may contribute to lower healthcare utilization and overall costs, supporting its cost-effectiveness. The proposed multidimensional program is designed for clinical settings and aligns with the typical duration of standard care, facilitating its future integration into routine practice. Its low implementation cost and compatibility with the structure of the National Health System make it a financially sustainable option. Additionally, the program encourages multidisciplinary collaboration. Including fitness assessments could also support clinical monitoring, and there is increasing demand for exercise-based interventions that specifically target strength improvements in this patient population.

### 5.1. Strengths

This protocol advocates for a multidimensional approach (supervised exercise and mindfulness) as a key strategy to optimize pain, disability, muscular fitness, quality of life, mental health, and gait parameters in individuals with CPLBP, aiming to achieve clinically relevant improvements. A notable strength of this study design is its use of a multidimensional approach with an active control group for treating CPLBP. Additionally, various measures will be used to assess both the physical and mental health status of the individuals, allowing for a comprehensive evaluation of the intervention’s potential effects. Fitness will be assessed through field tests, which, if proven beneficial, could have significant implications in clinical practice. Furthermore, the study will use objective measures (triaxial accelerometers) to examine associations between PA and sedentary behavior in this population. In addition, several potential challenges have been anticipated. Participant dropouts, particularly during the short-term follow-up period, will be addressed through regular contact, reminder communications, and flexible scheduling of assessments. To promote adherence to the intervention protocols, all sessions will be supervised, and attendance will be monitored. Variability among therapists will be minimized through standardized training and the use of detailed intervention protocols. Finally, while the study population may limit generalizability, the inclusion criteria were carefully selected to reflect real-world clinical populations with CPLBP.

### 5.2. Limitations

Despite its potential, this protocol presents several potential limitations. First, the study design does not include a control group without treatment due to ethical considerations related to patient care in the clinical setting. The study recognizes the potential risk of participant fatigue due to using multiple assessment instruments. To mitigate this, assessments will be scheduled with sufficient time, short breaks will be provided as needed, and participants will be instructed to respond carefully. Additionally, the chosen instruments are widely used and validated in clinical research involving chronic pain populations, ensuring their relevance and feasibility. Due to limited human resources, it will not be possible to implement blinded assessment in this study. Specifically, the same team members responsible for the intervention delivery will also be involved in outcome assessment, making it unfeasible to separate these roles. This represents a potential source of performance and detection bias. Although this limitation cannot be fully mitigated, we will apply standardized procedures for data collection and outcome assessment to minimize bias and ensure reliability. To enhance internal validity, future trials should aim for blind assessors where feasible and recruit larger and more diverse samples to increase generalizability. Furthermore, upcoming research should prioritize the addition of a no-treatment control group to clearly isolate the specific effects of combined mindfulness and exercise interventions.

### 5.3. Post-Trial Care and Dissemination Policy

The interventions involved in this trial are expected to cause no or minimal harm. No harm or compensation is anticipated for participants enrolled in the trial. The results obtained will be disseminated in the form of scientific papers, presentations at national and international conferences, as well as knowledge transfer activities for healthcare professionals.

## Figures and Tables

**Figure 1 healthcare-13-02065-f001:**
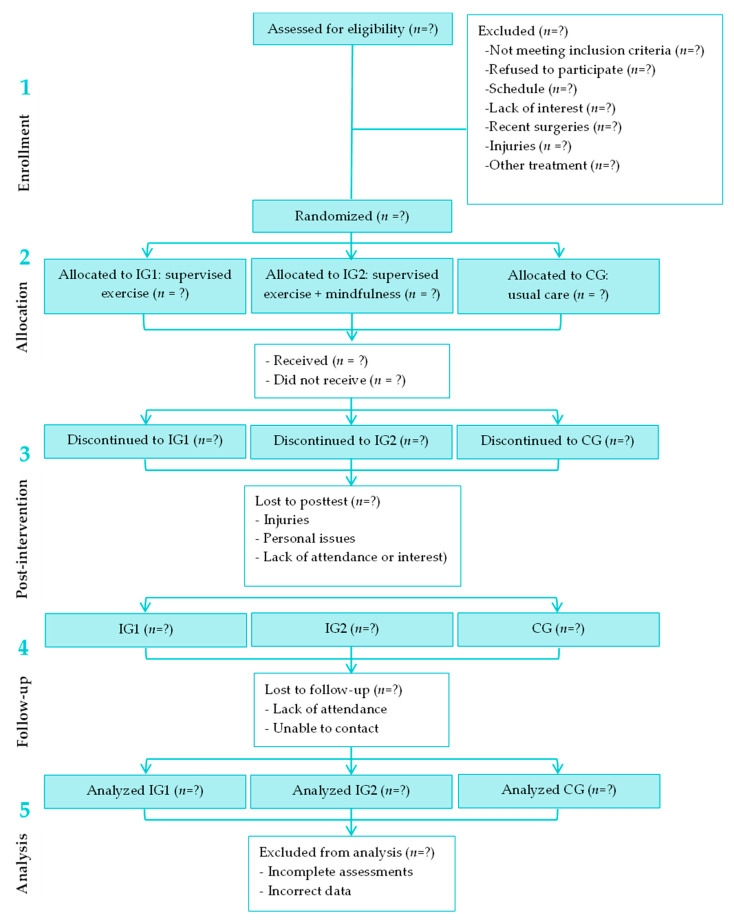
CONSORT diagram.

**Table 1 healthcare-13-02065-t001:** Recommended protocol items scheduled for enrolment, interventions, and assessments according to evidence [38].

	Enrolment	Baseline	Allocation	Intervention	After Intervention	Follow-Up
Timepoint	−**t_1_**	**t_0_**	**0**	**0**	**t_1_**	**t_2_**
Enrolment						
Preliminary contact	X					
Informative session	X					
Informed consent	X					
Randomization			X			
Allocation			X			
Interventions						
Supervised exercise				X		
Supervised exercise + mindfulness				X		
Control group				X		
Assessments						
Clinical information survey		X				X
C-AAF		X				
Blood pressure		X			X	X
Resting heart rate		X			X	X
Body composition		X			X	X
Pain related measures						
Pain threshold (algometry)		X			X	X
Perceived acute pain (VAS, NRS)		X		X	X	X
Pain catastrophizing (PCS)		X			X	X
Disability due to pain (ODI)		X			X	X
Central Sensitization (CSI)		X			X	X
Physical fitness tests		X			X	X
Biering–Sørensen		X			X	X
Prone Bridging		X			X	X
30-sec chair stand		X			X	X
Hand dynamometry		X			X	X
8-foot up-and-go		X			X	X
Gait parameters		X			X	X
Accelerometry		X			X	X
Self-reported sedentary behavior (SBQ-S)		X			X	X
Quality of life and mental health						
Health-related quality of life (SF-36)		X			X	X
Depression (BDI-II)		X			X	X
Anxiety (STAI)		X			X	X
Sleep quality (PSQI)		X			X	X
Rate of perceived exertion (RPE)		X		X	X	X

Abbreviation: −t_1_: screening phase, t_0_: Baseline, t_1_: post-test, t_2_: 2-month follow-up, BDI-II: Beck Depression Inventory-II, C-AAF: The Spanish version of the PAR-Q (Physical Activity Readiness Questionnaire), CSI: Central Sensitization Inventory, NRS: Numerical Rating Scale, ODI: Oswestry Disability Index, PCS: Pain Catastrophizing Scale, PSQI: Pittsburgh Sleep Quality Index, SBQ-S: Spanish version of the Sedentary Behavior Questionnaire, SF-36: 36-item Short-Form Health Survey, STAI: State-Trait Anxiety Inventory, VAS: Visual Analogue Scale.

**Table 2 healthcare-13-02065-t002:** Inclusion and exclusion criteria.

**Inclusion Criteria**
- Be previously diagnosed with CPLBP pain by a healthcare professional.
- Aged between 18 and 65 years old.
- Able to read and understand the informed consent, as well as the objective of the study.
- Able to walk and move independently.
- Able to communicate possible problems emerging during the evaluation tests.
**Exclusion Criteria**
- Serious structural disorders of the lumbar spine (e.g., spondylolysis, spondylolisthesis, canal stenosis, degenerative disc disease, disc herniation, tumour, trauma or fracture of the lumbar spine or lower limbs), cauda equina syndrome, and radicular leg pain of neuropathic origin.
- **Acute or terminal illness.**
- **Physical injury.**
- **Physical or mental illness.**
- **Medical conditions or treatments that contraindicate participation.**

Note: Individuals with referred or radiating pain (e.g., gluteal or posterior thigh), without neurological deficits or signs of nerve root involvement, will be considered eligible according to the current definition of chronic primary low back pain (CPLBP).

**Table 3 healthcare-13-02065-t003:** Structure of the intervention program by group and phase.

	Phase 1		Phase 2		Phase 3		Phase 4	
	Week 1	Week 2	Week 3	Week 4	Week 5	Week 6	Week 7	Week 8
	Session 1/2	Session 3/4	Session 5/6	Session 7/8	Session 9/10	Session 11/12	Session 13/14	Session 15/16
Equipment	Mats		Mats, Resistance Bands		Mats, 2-kg Dumbbells		Mats, Foam Rollers, Stability Balls and 2-kg Dumbbells	
IG1	5′ warm-up		5′ warm-up		5′ warm-up		5′ warm-up	
	35′ mobility, isometric and motor control exercises		35′ co-contraction exercises		35′ functional exercises with load		35′ functional exercises in unstable surface	
	5′ cool-down		5′ cool-down		5′ cool-down		5′ cool-down	
IG2	5′ warm-up		5′ warm-up		5′ warm-up		5′ warm-up	
	35′ mobility, isometric and motor control exercises		35′ co-contractions exercises		35′ functional exercises with load		35′ functional exercises on unstable surface	
	5′ cool-down		5′ cool-down		5′ cool-down		5′ cool-down	
	+2.5-h weekly mindfulness session							
	Sections: topic presentation, group dialogue, exploration and mindfulness practice							
CG	45′ of stretching, breathing and motor control exercises							
	Warm up: postural awareness, diaphragmatic and costal breathing, transversus abdominis contraction							
	Main activities: inferior and superior abdominal, hip abduction with knee extension, cat-camel and bird-dog exercises							
	Stretching: lumbosacral, psoas, hamstring and pyramidal muscle stretching in sitting and supine position							

**Table 4 healthcare-13-02065-t004:** Details of the resistance exercise program.

**PHASE 1 (≥5 RPE)**													
Warm-up	1. Lumbo-pelvic movement with breathing control; 2. Transversus abdominis activation; 3. Plank on the wall; 4. Hip rotation; 5. Floor slides with foam roller decubitus and lateral position)												
Main part			**1st session**			**2nd session**		**3rd session**					
	N		D		E	D	E	D				E	
	1		Good morning		Mat	The lunge	Mat	Static wall squat				Mat	
	2		Lumbo-pelvic movement in sitting position		Mat	Tie your shoelaces	Mat	Good morning				Mat	
	3		Left lifting progression		Mat	Adapted crunch	Mat	Monster Walk initiation				Mat	
	4		The cat		Mat	Adapted hollow rock	Mat	Bird dog initiation				Mat	
	5		Sun salute		Mat	Open yourself “like a book”	Mat	Adapted crunch				Mat	
	6		Lateral greeting		Mat	Lying on your side	Mat	Reversed bird dog				Mat	
	7		Bird dog initiation		Mat	Modified side plank	Mat	Glute bridge				Mat	
	8		-		-	-	-	Modified side plank				Mat	
Cool-down	Stretching of the dorsal, abdominal, hamstrings, oblique and psoas muscles												
**PHASE 2 (≥5 RPE)**													
Warm-up	1. Lumbo-pelvic movement with breathing control; 2. Transversus abdominis activation; 3. Plank on the wall; 4. Hip rotation; 5. Floor slides with foam roller decubitus and lateral position												
Main part			**1st session**			**2nd session**		**3rd session**				**4th session**	
			D		E	D	E	D		E		D	E
	1		Stading kick back		-	Deep breathing	Mat	Upper body mobility		Foam roller		Window washing at the wall	-
	2		Adapted flamingo		-	Glute bridge	Mat	Lunge on the floor		Mat		Draw crosses at the wall	-
	3		Adapted bird dog		-	Prone plank with leg raises	Mat	Lateral greeting		Mat		Clamshell	Resistance band
	4		Hedgehog		Stability ball	Bird dog initiation	Mat	Arms and legs coordination		Mat		Leg lifts	Mat
	5		Shoulder movement		Mat	Side plank rotations	Mat	Hypopressives in supine position		Mat		Leg opening	Mat
	6		Lying on the beach		Mat	Wall climb	-	Unilateral leg lifts		Mat		Bicycle	Mat
	7		Child pose		Mat	Catching object	-	Clamshell		Mat		Side Lying Hip Adduction	Mat
	8		Side hug		Mat	Lying crab	-	Horse kick		Mat		Triceps extension	Chair
Cool-down	Stretching of the dorsal, abdominal, hamstrings, oblique and psoas muscles												
**PHASE 3 (≥6 RPE)**													
Warm-up	1. Lumbo-pelvic movement with breathing control; 2. Transversus abdominis activation; 3. Plank on the wall; 4. Lateral greeting; 5. Bird-dog; 6. Adapted frog pose; 7. Floor slides with foam roller (decubitus and lateral position)												
Main part			**1st session**			**2nd session**		**3rd session**				**4th session**	
	N		D		E	D	E	D		E		D	E
	1		Activation of the spinal, hamstring and abdominal muscles		Mat, dumbbell	Activation of the spinal, hamstring and abdominal muscles	Mat, dumbbell	Dumbbell plank pull through (sliding)		Mat, dumbbell		Reverse crunch	Stick
	2		Glute bridge		Mat, dumbbell	Infinity	Dumbbell	Infinity		Dumbbell		Glute bridge	Mat, dumbbell
	3		Bird-dog		Mat, dumbbell	Stride	Dumbbell	Shoulder flexion		Dumbbell		Adapted shoulder tap	Mat
	4		Squat		Dumbbell	Squat with shoulder flexion	Dumbbell	Squat		Resistance band		Crawling	-
	5		Squats with shoulder flexion		Dumbbell	Dead bug	Mat, dumbbell	Starfish		Mat		Squat	Dumbbell
	6		Shoulder flexions		Dumbbell	Glute bridge	Mat, dumbbell	Activation of the spinal, hamstring and abdominal muscles		Mat		Shoulder and hip mobility	Mat
	7		Dumbbell one		Dumbbell	Window washing with feet	Mat	Dumbbell squats		Dumbbell		Dumbbell squats	Dumbbell
	8		Half squat		Resistance band	Swing	Mat, dumbbell	Adapted stair climbing		Mat		Adapted stair climbing	Mat
Cool-down	1.Stretching of the dorsal, abdominal, hamstrings, oblique and psoas muscles												
**PHASE 4 (≥6 RPE)**													
Warm-up	1. Lumbo-pelvic movement with breathing control; 2. Transversus abdominis activation; 3. Adapted skipping; 4. Adapted Jumping Jack; 5. Plank on the floor; 6. Adapted mountain climber at the wall; 7. Floor slides with foam roller (decubitus and lateral position)												
Main part	**1st session**				**2nd session**		**3rd session**		**4th session**			**5th session**	
	N	D		E	D	E	D	E	D		E	D	E
	1	Mountain climber		Mat	Side steps	Mat	Bird-dog	Stability ball	Planks		Stability ball	Crunches	Stability ball
	2	Single-leg balance with arms extended		Mat	Standing lateral crunches	Mat	Mountain climber	Mat	Raising legs		Stability ball	Bird-dog	Stability ball
	3	Balance drill		Mat	Bulgarian squats	Stability ball	Standing lateral crunches	Mat	Hip thrust		Stability ball	Adapted shoulder tap	Balance trainer
	4	Raising legs		Stability ball	Raising legs	Stability ball	Side steps with overhead press	Balance trainer, dumbbell	Bird-dog		Stability ball	Single-leg balance with arms extended	Mat
	5	Hip anteversion and retroversion		Stability ball	Bird-dog	Stability ball	Raising legs	Stability ball	Adapted skipping		Balance trainer	Standing lateral crunches	Mat
	6	Initiation to skipping		Balance trainer	Side steps	Balance trainer	Squats	Balance trainer	Bulgarian squats		Balance trainer	Side steps with overhead press	Balance trainer, dumbbell
	7	Squats		Balance trainer	Plank leg raises	Balance trainer	Adapted skipping	Balance trainer	Squats		Balance trainer	Squats	Balance trainer
Cool-down	Stretching of the dorsal, abdominal, hamstrings, oblique and psoas muscles												

Abbreviation: Equip.: equipment, RPE: Rate of perceived exertion.

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
