# Peer review of "Effects of a Multidimensional Exercise and Mindfulness Approach Targeting Physical, Psychological, and Functional Outcomes: Protocol for the BACKFIT Randomized Controlled Trial with an Active Control Group"

_healthcare, 2025, doi:10.3390/healthcare13162065_

Round 1
Reviewer 1 Report
Comments and Suggestions for Authors
Dear authors,
Thank you for give me the possibility to revise your work “ Effects of a multidimensional approach based on exercise and mindfulness on pain, disability, quality of life, mental health, and gait parameters in patients with chronic primary low back pain: the BACKFIT randomized controlled trial protocol.”.
After reviewing your paper, here are a few suggestions to improve your paper.
Abstract
- I would add more statistical results in order to make your abstract more attractive.
- Please be sure that your keyword are MeSh terms.
Introduction
- The burden of low back pain should be better described in order to support your work, please use the lancet series of low back pain, especially the first (doi: 10.1016/S0140-6736(21)00733-9.)
- I would better described the research gap that the study aims to fill, especially in the context of previous study
- Lines 81-83 trunk exercise. I suggest to add a references concerning the impornance of trunk muscles in low back pain doi: 10.1080/10669817.2023.2252202.
METHOD
- The study is a RCT, please add the number of clinical trail registration
- When was the study conducted?
- Details on the blinding process are limited. Please expand this aspect in order to minimize bias and clarify the study's validity.
- How was the sample size chosen?
- Inclusion criteria: “Be previously diagnosed with CPLBP pain by a healthcare professional” I think that there are more differences in CPLBP assessments from a physiotherapist respect other healthcare professional. Please justify and specify who assessed these patients.
Results
- I suggest to improve the quality of your tables. For example to use only three lines for your tables. It make it more elegant.
Discussion
- There are some abbreviations that make the test difficult to read. I suggest to use the full name in the first paragraph.
- I suggest to summarize better your main findings in the first paragraph.
- I suggest a more comprehensive comparation with previous studies
- Expanding on the potential mechanisms underlying of your results could deepen understanding and suggest avenues for further investigation
- I would add some limitation in order to provide a more balanced view
Author Response
Effects of a multidimensional approach based on exercise and mindfulness on pain, disability, quality of life, mental health, and gait parameters in patients with chronic primary low back pain: the BACKFIT randomized controlled trial protocol.
Comments 1: Abstract. I would add more statistical results to make your abstract more attractive.
|
|||||||||||||||||||||||||||||||||
Response 1: Done. “It will include per-protocol and intention-to-treat approaches (≥70% of attendance), with program effects assessed via one-way ANCOVA for between-group changes in primary and secondary outcomes”. |
|||||||||||||||||||||||||||||||||
Comments 2: Abstract. Please be sure that your keyword are MeSh terms. |
|||||||||||||||||||||||||||||||||
Response 2: Done. Thank you.
Comments 3: The burden of low back pain should be better described in order to support your work, please use the lancet series of low back pain, especially the first (doi: 10.1016/S0140-6736(21)00733-9.). Response 3: We thank the reviewer for this helpful suggestion. In response, we have expanded the section describing the global burden of low back pain in the Introduction. To strengthen the clinical and public health relevance of our study, we have incorporated three key references: the first two articles of The Lancet Low Back Pain Series (Hartvigsen et al., 2018; Foster et al., 2018), and a more recent comprehensive review on the topic (Knezevic et al., 2021). Together, these references support the increasing recognition of LBP as a major global health challenge and help contextualize the importance of developing targeted management strategies for CPLBP. The corresponding change has been made in the revised manuscript and can be found in lines 50-66.
Comments 4: I would better describe the research gap that the study aims to fill, especially in the context of previous study. Response 4: We thank the reviewer for this helpful comment. In response, we have substantially revised the Introduction section to more clearly articulate the research gap that this study aims to address, particularly in relation to previous work in the field. We believe the new version provides a more comprehensive and focused justification for the study’s objectives and design (lines 50-164).
Comments 5: Lines 81-83 trunk exercise. I suggest to add a references concerning the importance of trunk muscles in low back pain doi: 10.1080/10669817.2023.2252202 Response 5: We thank the reviewer for this helpful suggestion. We have now included the reference by Deodato et al. (2024), which provides evidence of trunk muscle impairments associated with chronic low back pain in adolescent gymnasts. Although the population in that study differs from ours, we consider these findings relevant to emphasize the clinical role of trunk musculature in the context of low back pain. The reference has been added in the revised paragraph (lines 102-106). Comments 6: The study is a RCT, please add the number of clinical trail registration Trial registration. Response 6: Done. (line 48). Comments 7: When was the study conducted?, Response 7: We thank the reviewer for the helpful comment. The study was conducted in March 2022. Comments 8: Details on the blinding process are limited. Please expand this aspect in order to minimize bias and clarify the study's validity Response 8: Done. Please, see it in the Limitation section (pages 707-712).
Comments 9: How was the sample size chosen? Response 9: We appreciate your observation. However, this point is already addressed in the manuscript [participant´s section]. We respectfully refer you to that part for clarification. Based on previous literature (Owen et al., 2020) with a statistical power of 0.80, an alpha error of 0.05, and an effect size of 0.70, a sample size of 26 participants per group is sufficient to find differences in pain after the intervention. Considering dropouts rate of approximately 20-30%, as reported in similar studies involving pain populations (Kamper et al., 2015), the total sample size will be set at 105 participants, randomly assigned to intervention group 1 (IG1, n=35), intervention group 2 (IG2, n=35) and control group (CG, n=35).
Comments 10: Inclusion criteria: “Be previously diagnosed with CPLBP pain by a healthcare professional” I think that there are more differences in CPLBP assessments from a physiotherapist respect other healthcare professional. Please justify and specify who assessed these patients. Response 11: We thank the reviewer for this relevant observation. We have clarified in the manuscript that the diagnosis of CPLBP will be performed and/or confirmed by medical specialists in Physical and Rehabilitation Medicine, all of whom are experienced in the assessment and management of musculoskeletal disorders (lines 253-255).
Comments 12: I suggest to improve the quality of your tables. For example, to use only three lines for your tables. It makes it more elegant. Response 12: Thank you for pointing this out. Done.
Comments 13: There are some abbreviations that make the text difficult to read. I suggest to use the full name in the first paragraph. Response 13: Done.
Comments 14: I suggest to summarize better your main findings in the first paragraph. Response 14: We appreciate the reviewer’s suggestion. In response, we have revised the first paragraph of the Discussion to provide a more concise and focused summary of the main findings, ensuring they are clearly presented at the beginning of the section (lines 592-617).
Comments 15: I suggest a more comprehensive comparation with previous studies Response 15: Done. We would like to thank the reviewer for their insightful feedback regarding the Discussion section. In response to this suggestion, we have substantially revised and rewritten the entire section to provide a more focused, coherent, and in-depth interpretation of the findings and their clinical implications. We believe that these changes significantly strengthen the manuscript and better align the discussion with the study's objectives and expected outcomes.
Comments 16: Expanding on the potential mechanisms underlying of your results could deepen understanding and suggest avenues for further investigation. Response 16: Thank you for this valuable suggestion. In response, we have included a dedicated paragraph in the Discussion section addressing the expected physiological mechanisms that may underline the effects of the combined intervention. These mechanisms include neuroplastic changes, reductions in central sensitization, and improvements in motor control and proprioception, all of which are supported by existing literature on chronic pain and physical exercise lines (607-631).
Comments 17: I would add some limitation in order to provide a more balanced view Response 17: Done. We have included this paragraph to provide a more balanced view in the limitations section (lines 696-709). |

Reviewer 2 Report
Comments and Suggestions for Authors
This article outlines a promising study protocol that addresses a significant health issue. By implementing the suggestions for each section, the authors can enhance the clarity, depth, and impact of their work. Engaging with more recent literature and emphasizing implications for practice will strengthen the article further.

Author Response

(The authors gave the same response as above.)

Reviewer 3 Report
Comments and Suggestions for Authors
Reviewer's comments
The protocol addresses a topic of high clinical relevance: the treatment of primary chronic low back pain (CPLBP) by means of a multidimensional approach. The trial design is rigorous and detailed, following guidelines such as SPIRIT and CONSORT. However, there are substantive areas that require greater clarity, justification or reformulation.
Abstract
- It is recommended to improve the wording of the objective to make it clearer and more direct.
- Explicitly add that this is a registered protocol.
- A justification or introduction must be provided.
Introduction
- A differentiation should be made between unspecific and specific chronic low back pain. Lack of clearer delimitation between non-specific low back pain (NSCLBP) and CPLBP.
- The research problem should be better justified and research gaps should be specified. Further elaborate on the evidence gaps that justify a new RCT, beyond the potential usefulness of the multidimensional approach.
- The authors should explain what supervised versus unsupervised exercise consists of. Is guided and unguided the same? Is the therapist present? Telerehabilitation? It should be clarified
- Include criticisms of previous studies with methodological limitations (lack of follow-up, not including mindfulness, etc.).
Methods
- Justify the use of ANCOVA instead of mixed models, especially since there is 2-year follow-up.
- Explicitly include methods to minimize bias: blinding of analysts, control of the expectation effect, etc.
- The intervention in the control group (CG) is similar to an active program - it does not represent "passive standard care". Reconsider the design of the control group or justify its inclusion as an active control. Must be justified
- It is not made clear whether the mindfulness instructor will participate in other phases (evaluations, for example), which may introduce biases.
- The methods for controlling type I errors (multiple comparisons) are not detailed. Consider including statistical adjustment plans for multiplicity (e.g., Bonferroni ).
- which health professional performs the diagnosis?
- Subjects with neuropathic pain are excluded? radiating pain? referred pain to the leg?
- The most important thing is the dosage of the exercises. Time, repetitions, speed, progression.... this must be justified. The authors determine that the progression will be individualized and may be a bias... will some subjects perform fewer repetitions? The time is the same for all? This should be clarified.
- Will the exercises be in order of progression by difficulty or randomly?
- We know that the contraction of the transverse abdominis is complicated and needs to be learned. How is this established in patients? How do they know if they are contracting that muscle?
Results
- I propose to the authors to reorganize the measurement variables by grouping them by clinical domains (e.g., “Pain and disability”, “Physical condition”, “Mental health”, etc.).
- Include expected MCID values for Oswestry, PCS and other key scales to facilitate future clinical interpretation.
- Justify the selection of each instrument or eliminate the less sensitive ones if there is conceptual overlap.
- To better detail the justification of psychological tools (e.g. STAI, BDI-II) in the mindfulness framework.
- variables collected with the accelerometer must be specified
Discussion
- Include a section on expected physiological mechanisms (e.g., neuroplasticity, reduced central sensitization, improved motor control).
- Compare specifically with previous similar studies (e.g., with exercise + education, or exercise + mindfulness in other pathologies).
- It mentions the lack of blinding and the impossibility of a placebo group, but without suggesting ways to mitigate this. Propose possible solutions: use blinded assessors, record patient expectations, sensitivity analysis by adherence group.
- The use of multiple instruments is presented as a strength, but can be a weakness if there is fatigue or neglect.
Author Response

(The authors gave the same response as above.)

Round 2
Reviewer 3 Report
Comments and Suggestions for Authors
Most of the reviewer's suggestions and contributions have been addressed. Thank you very much.
The registration number should be inside the abstract.
Author Response
Reviewer #1
Comment
Most of the reviewer's suggestions and contributions have been addressed.
The registration number should be inside the abstract.
Response
Thank you very much for your suggestion. The trial registration number (NCT05443880) has been added to the abstract, as requested.
